# CtRL-Sim: Reactive and Controllable Driving Agents with Offline Reinforcement Learning

**Luke Rowe**[*1,2], **Roger Girgis**[*1,3,6], **Anthony Gosselin**[1,3], **Bruno Carrez**[1],
**Florian Golemo**[1], **Felix Heide**[4,6], **Liam Paull**[1,2,5], **Christopher Pal**[1,3,5]
[1]Mila, [2]Université de Montréal, [3]Polytechnique Montréal, [4]Princeton University,
[5]CIFAR AI Chair, [6]Torc Robotics
https://montrealrobotics.ca/ctrlsim

**Abstract:** Evaluating autonomous vehicle stacks (AVs) in simulation typically involves replaying driving logs from real-world recorded traffic. However, agents replayed from offline data are not reactive and hard to intuitively control. Existing approaches address these challenges by proposing methods that rely on heuristics or generative models of real-world data but these approaches either lack realism or necessitate costly iterative sampling procedures to control the generated behaviours. In this work, we take an alternative approach and propose CtRL-Sim, a method that leverages return-conditioned offline reinforcement learning (RL) to efficiently generate reactive and controllable traffic agents. Specifically, we process real-world driving data through a physics-enhanced Nocturne simulator to generate a diverse offline RL dataset, annotated with various rewards. With this dataset, we train a return-conditioned multi-agent behaviour model that allows for fine-grained manipulation of agent behaviours by modifying the desired returns for the various reward components. This capability enables the generation of a wide range of driving behaviours beyond the scope of the initial dataset, including adversarial behaviours. We show that CtRL-Sim can generate realistic safety-critical scenarios while providing fine-grained control over agent behaviours.

**Keywords:** Autonomous Driving, Simulation, Offline Reinforcement Learning

## 1 Introduction

Recent advances in autonomous driving has enhanced their ability to safely navigate the complexities of urban driving [1]. Despite this progress, ensuring operational safety in long-tail scenarios, such as unexpected pedestrian behaviours and distracted driving, remains a significant barrier to widespread adoption. Simulation has emerged as a promising tool for efficiently validating the safety of autonomous vehicles (AVs) in these long-tail scenarios. However, a core challenge in developing a simulator for AVs is the need for other agents within the simulation to exhibit realistic and diverse behaviours that are reactive to the AV, while being easily controllable. The traditional approach for evaluating AVs in simulation involves fixing the behaviour of agents to the behaviours exhibited in pre-recorded driving data. However, this testing approach does not allow the other agents to react to the AV, which yields unrealistic interactions between the AV and the other agents.

To address the issues inherent in non-reactive log-replay testing, prior work has proposed rule-based methods [2, 3] to enable reactive agents. However, the behaviour of these rule-based agents often lacks diversity and is unrealistic. More recently, generative models learned from real-world data have been proposed to enhance the realism of simulated agent behaviours [4, 5, 6, 7, 8, 9]. While these methods produce more realistic behaviours, they are either not easily controllable [4, 5, 9] or require costly sampling procedures to control the agent behaviours [10, 8, 7, 11, 12].

---

[*]Denotes equal contribution. Corresponding email: luke.rowe@mila.quebec

8th Conference on Robot Learning (CoRL 2024), Munich, Germany.

In this paper, we propose CtRL-Sim to address these limitations of prior work. The CtRL-Sim framework utilizes return-conditioned offline reinforcement learning (RL) to enable reactive, *closed-loop*, controllable, and probabilistic behaviour simulation within a physics-enhanced Nocturne [13] environment. We process scenes from the Waymo Open Motion Dataset [14] through Nocturne to curate an offline RL dataset for training that is annotated with reward terms such as "vehicle-vehicle collision" and "goal achieved". We propose a return-conditioned multi-agent autoregressive Transformer architecture [15] within the CtRL-Sim framework to imitate the driving behaviours in the curated dataset. We then leverage exponential tilting of the predicted return distribution [16] as a simple yet effective mechanism to control the simulated agent behaviours. While [16] exponentially tilts towards more optimal outcomes for the task of reward-maximizing control, we instead propose to tilt in *either direction* to provide control over both good and bad simulated driving behaviours.

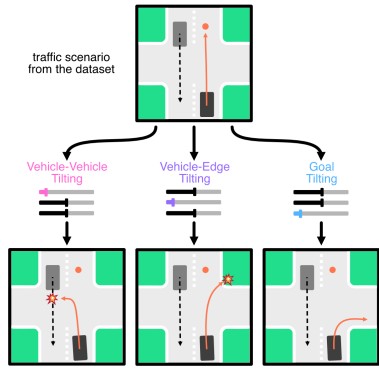

Figure 1: **CtRL-Sim allows for controllable agent behaviour** from existing datasets. This allows users to create interesting edge cases for testing and evaluating AV planners.

We show examples of how CtRL-Sim can be used to generate counterfactual scenes when exponentially tilting the different reward axes in Figure 1. For controllable generation, CtRL-Sim simply requires specifying a tilting coefficient along each reward axis, which circumvents the costly iterative sampling required by prior methods. CtRL-Sim scenarios are simulated within our physics-extended Nocturne environment. We summarize our main contributions: **1.** We propose CtRL-Sim, which is, to the best of our knowledge, the first framework applying return-conditioned offline RL for controllable and reactive behaviour simulation. Specifically, CtRL-Sim employs exponential tilting of factorized reward-to-go to control different axes of agent behaviours. **2.** We propose an autoregressive multi-agent encoder-decoder Transformer architecture within the CtRL-Sim framework that is tailored for controllable behaviour simulation. **3.** We extend the Nocturne simulator [13] with a Box2D physics engine, which facilitates realistic vehicle dynamics and collision interactions.

We demonstrate the effectiveness of CtRL-Sim at producing controllable and realistic agent behaviours compared to prior methods. We also show that finetuning our model in Nocturne with simulated adversarial scenarios enhances control over adversarial behaviours. CtRL-Sim has the potential to serve as a useful framework for enhancing the safety and robustness of AV planner policies through simulation-based training and evaluation.

## 2 CtRL-Sim

In this section, we present the proposed CtRL-Sim framework for behaviour simulation. We first introduce CtRL-Sim in the single-agent setting, and subsequently show how it extends to the multi-agent setting. Given the state of an agent $s_t$ at timestep $t$ and additional context (e.g., the road structure, the agent's goal), the behaviour simulation model employs a driving policy $\pi(a_t|s_t, m, s_G)$ and a forward transition model $\mathcal{P}(s_{t+1}|s_t, a_t)$ to control the agent in the scene. Note that $a_t$ is the action, $m$ is the map context, and $s_G$ is the prescribed goal state. Using the physics-extended Nocturne simulator, we have access to a physically-realistic forward transition model $\mathcal{P}$. In this work, we are interested in modelling the policy $\pi(a_t|s_t, m, s_G)$ such that we can both imitate the real distribution of driving behaviour and control the agent's behavior to generate long-tail counterfactual scenes.

### 2.1 Our Approach to Controllable Simulation via Offline RL

We consider the common offline RL setup where we are given a dataset $\mathcal{D}$ of trajectories $\tau_i = \{\ldots, s_t, a_t, r_t, \ldots\}$, with states $s_t \in \mathcal{S}$, actions $a_t \in \mathcal{A}$ and rewards $r_t$. These trajectories are generated using a (suboptimal) behaviour policy $\pi_B(a_t|s_t)$ executed in a finite-horizon Markov

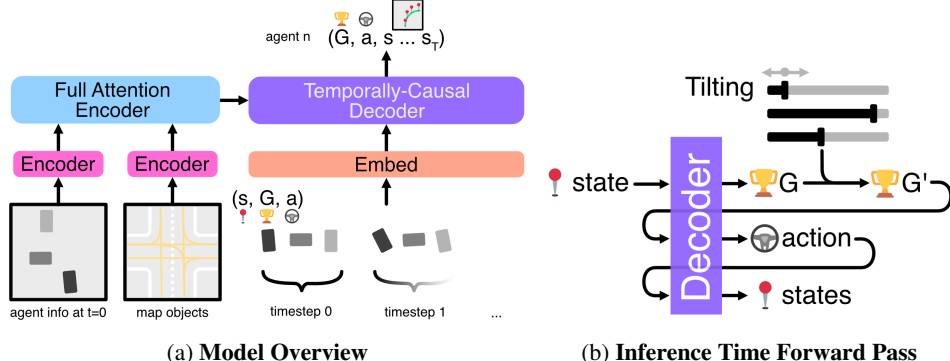

(a) **Model Overview**  (b) **Inference Time Forward Pass**

Figure 2: 2a (left) The agent and map data at $t = 0$ are encoded and fed through a Transformer encoder as context for the decoder, similar to [9]. Trajectories are arranged first by agents, then by timesteps, embedded, and fed through the decoder. For each agent, we encode $(s_t, G_t, a_t)$ (i.e. state, return-to-go, action) and we predict from these $(G_t, a_t, s_{t+1}, \ldots, s_T)$. 2b (right) At inference time, the state predicts the return-to-go. The return-to-go is tilted (i.e., reweighed to encourage specific behaviors) and is used to predict the action, which in turn is used to predict the next states.

decision process. The return-to-go at timestep $t$ is defined as the cumulative sum of scalar rewards obtained in the trajectory from timestep $t$, $G_t = \sum_{t'=t}^{T} r_{t'}$. The objective of offline RL is to learn policies that perform as well as or better than the best agent behaviours observed in $\mathcal{D}$.

The primary insight of this work is the observation that offline RL can be an effective way to perform controllable simulation. That is, the policy distribution over actions can be tilted at inference time towards desirable or undesirable behaviors by specifying different values of return-to-go $G_t$. This requires a different formulation of the policy such that it is conditioned on the return $\pi(a_t|s_t, G_t, s_G)^2$. In Table 3, we outline how different approaches in offline RL have learned return-conditioned policies. In this work, we adopt an approach that learns the joint distribution of returns and actions of an agent in a given dataset. Specifically, $p_\theta(a_t, G_t|s_t, s_G) = \pi_\theta(a_t|s_t, s_G, G_t)p_\theta(G_t|s_t, s_G)$. We note that [17] found it helpful to also utilize a *model-based* return-conditioned policy, whereby the future state is modelled as part of the joint distribution being learned. This is shown to provide a useful regularizing signal for the policy, even though the future state prediction is not directly used at inference time. In this work, we also found it helpful to regularize the learned policy by predicting the full sequence of future states. The final distribution we are aiming to model is thus given by $p_\theta(s_{t+1:T}, a_t, G_t|s_t, s_G) = p_\theta(s_{t+1:T}|s_t, s_G, G_t, a_t)\pi_\theta(a_t|s_t, s_G, G_t)p_\theta(G_t|s_t, s_G)$.

At inference time, we obtain actions by first sampling returns $G_t \sim p_\theta(G_t|s_t, s_G)$ and then sampling actions $a_t \sim \pi_\theta(a_t|s_t, s_G, G_t)$. This sampling procedure corresponds to the imitative policy since the sampled returns are obtained from the learned density that models the data distribution. Following prior work in offline RL [16, 17, 18], we can also sample actions from an exponentially-tilted policy distribution. This is done by sampling the returns from the tilted distribution $G_t' \sim p_\theta(G_t|s_t, s_G) \exp(\kappa G_t)$, with $G_t'$ being the tilted return-to-go and where $\kappa$ represents the inverse temperature; higher values of $\kappa$ concentrate more density around the best outcomes or higher returns, while negative values of $\kappa$ concentrate on less favourable outcomes or lower returns.

We are interested in modelling and controlling the individual components of the reward function rather than maximizing their weighted sum. For example, we would like to model an agent's ability to reach its goal, drive on the road, and avoid collisions. In general, given $C$ reward components, our objective is to learn policies that are conditioned on *all* its factored dimensions as this would grant us control over each one at test time. This entails modelling separate return components as $G_t^c \sim p_\theta(G_t^c|s_t, s_G)$ for each return component $c$. Applying this factorization, we reformulate the learned policy to explicitly account for the conditioning on all return components $\pi_\theta(a_t|s_t, s_G, G_t^1, \ldots G_t^C)$. At test time, each return component will be accompanied by its own inverse temperature $\kappa^c$ to

---

[2]Note that we omit the additional context $m$ for brevity.

enable control over each return component, which enables sampling actions that adhere to different behaviours specified by $\{\kappa^1, \ldots, \kappa^C\}$, as shown in Algorithm 2 in Appendix B.

To implement our framework for behaviour simulation, we extend the approach presented above to the multi-agent setting. Across all agents we have sets for the joint states $\mathbb{S}_t$, goal states $\mathbb{S}_G$, actions $\mathbb{A}_t$, and returns-to-go $\mathbb{G}_t$. The final multi-agent joint distribution we model is:

$$p_\theta(\mathbb{S}_{t+1:T}, \mathbb{A}_t, \mathbb{G}_t | \mathbb{S}_t, \mathbb{S}_G) = p_\theta(\mathbb{S}_{t+1:T} | \mathbb{S}_t, \mathbb{S}_G, \mathbb{G}_t, \mathbb{A}_t)\pi_\theta(\mathbb{A}_t | \mathbb{S}_t, \mathbb{S}_G, \mathbb{G}_t)p_\theta(\mathbb{G}_t | \mathbb{S}_t, \mathbb{S}_G), \quad (1)$$

where the returns and actions from the previous timesteps are shared across agents, while at the present timestep they are masked out so one can only observe one's own return and action.

## 2.2 Multi-Agent Behaviour Simulation Architecture

In this section, we introduce the proposed architecture for multi-agent behaviour simulation within the CtRL-Sim framework that parameterizes the multi-agent joint distribution presented in Equation (1). We propose an encoder-decoder Transformer architecture [19], as illustrated in Figure 2, where the encoder encodes the initial scene and the decoder autoregressively generates the trajectory rollout for all agents in the scene.

**Encoder** To encode the initial scene, we first process the initial agent states and goals $(\mathbf{s}_0, \mathbf{s}_G)$ and the map context $m$, where $\mathbf{s}_0$ is the joint initial state of all agents and $\mathbf{s}_G$ is the joint goal state of all agents. Each agent $i$'s initial state information $s_0^i$, which includes the position, velocity, heading, and agent type, is encoded with an MLP. Similarly, each agent's goal $s_G^i$, which is represented as the ground-truth final position, velocity, and heading, is also encoded with an MLP. We then concatenate the initial state and goal embedding of each agent and embed them with a linear layer to get per-agent embeddings of size $d$. We additionally apply an additive learnable embedding to encode the agents' identities across the sequence of agent embeddings. The map context is encoded using a polyline map encoder, detailed more fully in Appendix F, which yields $L$ road segment embeddings of size $d$. The initial agent embeddings and road segment embeddings are then concatenated into a sequence of length $N + L$ and processed by a sequence of $E$ Transformer encoder blocks.

**Decoder** The proposed decoder architecture models the joint distribution in Equation (1) as a sequence modelling problem, where we model the probability of the next token in the sequence conditioned on all previous tokens $p_\theta(x_t | x_{<t})$ [15]. In this work, we consider trajectory sequences of the form: $x = \langle \ldots, (s_t^1, s_G^1), (G_t^{1,1}, \ldots, G_t^{C,1}), a_t^1, \ldots, (s_t^N, s_G^N), (G_t^{1,N}, \ldots, G_t^{C,N}), a_t^N, \ldots \rangle$. These sequences are an extension of the sequences considered in the Multi-Game Decision Transformer [16] to the multi-agent goal-conditioned setting with factorized returns. Unlike Decision Transformer [15], our model predicts the return distribution and samples from it at inference time, which enables flexible control over the agent behaviours and circumvents the need to specify an expert return-to-go. We obtain state-goal tuple $(s_t^i, s_G^i)$ embeddings in the same way that $(\mathbf{s}_0, \mathbf{s}_G)$ are processed in the encoder. Following recent work that tokenizes driving trajectories [20, 9], we discretize the actions and return-to-gos into uniformly quantized bins. We then embed the action and return-to-go tokens with a linear embedding. To each input token, we additionally add two learnable embeddings representing the agent identity and timestep, respectively. The tokenized sequence is then processed by $D$ Transformer decoder layers with a temporally causal mask that is modified to ensure that the model is permutation equivariant to the agent ordering (see Appendix F for details).

**Training** Given a dataset of offline trajectories (Section 3), we train our model by sampling sequences of length $H \times N \times 3$, where $H$ is the number of timesteps in the context. The state, return-to-go, and action token embeddings output by the decoder are used to predict the next return token, action token, and future state sequence, respectively. We train the return-to-go and action headers with the standard cross-entropy loss function and the future state sequence header with an L2 regression loss function. The final loss function is of the form: $\mathcal{L} = \mathcal{L}_{\text{action}} + \mathcal{L}_{\text{return-to-go}} + \alpha \mathcal{L}_{\text{state}}$.

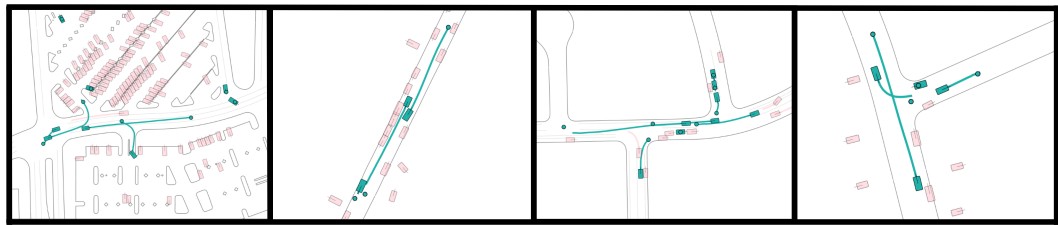

Figure 3: **Qualitative results of multi-agent simulation with CtRL-Sim.** The teal agents are controlled by CtRL-Sim, and other agents in pink are set to log-replay through physics.

| Method | ADE (m) ↓ | FDE (m) ↓ | Goal Success Rate (%) ↑ | JSD ($\times 10^{-2}$) ↓ | Collision (%) ↓ | Off Road (%) ↓ | Per Scene Gen. Time (s) ↓ |
|---|---|---|---|---|---|---|---|
| Replay-Physics* | 0.47 | 0.97 | 87.3 | 7.6 | 2.8 | 10.7 | 1.1 |
| Actions-Only [9] | 4.81±0.52 | 11.89±1.42 | 32.7±1.4 | 10.4±0.3 | 19.9±1.2 | 27.6±1.0 | 3.3 |
| Imitation Learning | 1.24±0.05 | 1.95±0.10 | 77.4±1.3 | 8.3±0.1 | 5.8±0.2 | 12.1±0.2 | 3.4 |
| DT (Max Return) [15] | 1.56±0.04 | 3.07±0.16 | 63.3±0.8 | 8.4±0.1 | 5.3±0.3 | 11.0±0.2 | 20.7 |
| CTG++† [11] | 1.73±0.10 | 4.02±0.32 | 38.8±5.4 | 7.4±0.2 | 5.9±0.4 | 15.0±1.5 | 44.0 |
| CtRL-Sim (No State Prediction) | 1.32±0.03 | 2.21±0.06 | 72.4±0.8 | 8.2±0.2 | 6.1±0.4 | 12.0±0.3 | |
| CtRL-Sim (Base) | 1.29±0.04 | 2.13±0.08 | 73.0±1.3 | 8.1±0.2 | 5.8±0.4 | 11.8±0.2 | 8.2 |
| CtRL-Sim (Positive Tilting) | 1.25±0.03 | 2.04±0.08 | 72.9±1.5 | 7.9±0.1 | 5.3±0.2 | 11.0±0.2 | |
| DT* (GT Initial Return) | 1.10±0.02 | 1.58±0.07 | 77.5±1.5 | 8.4±0.1 | 5.3±0.3 | 11.9±0.3 | 20.8 |
| CtRL-Sim* (GT Initial Return) | 1.09±0.02 | 1.60±0.06 | 77.2±1.1 | 8.1±0.2 | 5.6±0.4 | 12.2±0.1 | |

Table 1: **Multi-agent simulation results over 1000 test scenes.** We report mean±std across 5 seeds. CtRL-Sim achieves a good balance between reconstruction performance, common sense, realism, and efficiency. * indicates privileged models requiring GT future. † indicates reimplementation.

## 3 Experiments

### 3.1 Experimental Setup

**Offline RL Dataset** We curate an offline RL dataset derived from the Waymo Open Motion Dataset (WOMD) [14]. We extend Nocturne by integrating a physics engine based on the Box2D library for enabling realistic vehicle dynamics and collisions, detailed in Appendix C.1. Each scene in the Waymo dataset is fed through the physics-enhanced Nocturne simulator to compute the per-timestep actions and factored rewards for each agent. The factored rewards comprise of a *goal position*, *vehicle-vehicle collision*, and a *vehicle-road-edge collision* reward, detailed in Appendix C.2.

**Evaluation** We evaluate CtRL-Sim on its ability to replicate the driving behaviours found in the Waymo Open Motion Dataset (*imitation*) and generate counterfactual scenes that are consistent with specified tilting coefficients (*controllability*). For both modes of evaluation, we use 1 second of history and simulate an 8 second future rollout. For *imitation*, we evaluate on up to 8 moving agents per scene that we control with CtRL-Sim, where the remaining agents are set to log replay through physics. We evaluate on 1000 random test scenes in both modes of evaluation. Following recent work [7], we use three types of metrics for imitation evaluation: *reconstruction* metrics, such as Final Displacement Error (**FDE**), Average Displacement Error (**ADE**), and **Goal Success Rate**; a *distributional realism* metric (**JSD**) defined by the mean of the Jensen-Shannon Distances computed on linear speed, angular speed, acceleration, and distance to nearest vehicle features between real and simulated scenes; and *common sense* metrics measured by **Collision** and **Offroad** rate.

For controllability evaluation, we evaluate on 1 selected "interesting" interactive agent that is controlled by CtRL-Sim, defined as an agent who is moving and whose goal is within 10 metres of another moving agent. All agents except for the CtRL-Sim-controlled interesting agent are set to log replay through physics. We evaluate the model's controllability through metrics aligned with the specified reward dimensions: we report the goal success rate for the goal reward control, collision rate for the vehicle-vehicle reward control, and offroad rate for the vehicle-road-edge reward control.

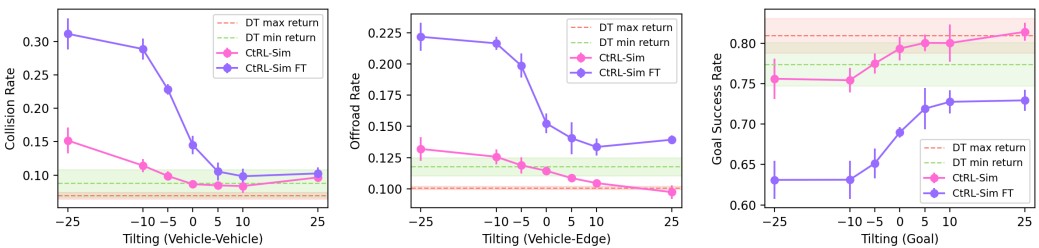

Figure 4: **Effects of exponential tilting.** Comparison of CtRL-Sim base model (magenta) and fine-tuned model (purple) across different reward dimensions. Rewards range from -25 to 25 for vehicle-vehicle collision (**left**), vehicle-edge collision (**middle**), and goal reaching (**right**). Results show smooth controllability, with fine-tuning enhancing this effect. We report mean±std over 5 seeds.

**Methods under Comparison** For imitation evaluation, we compare CtRL-Sim against several relevant baselines: **1.** *Replay-Physics* employs an inverse bicycle model to obtain the ground-truth log-replay actions and executes through the simulator. **2.** *Actions-Only* is an encoder-decoder model inspired by [9] where the decoder trajectory sequences only contain actions. **3.** *Imitation Learning (IL)* is identical to the architecture in Section 2.2 except with the removal of returns and the future state prediction. **4.** *Decision Transformer (DT)*: The *GT Initial Return* variant specifies the initial ground-truth return-to-go from the offline RL dataset, with the goal of acting as an imitative policy. *Max Return* follows the standard DT approach of selecting the maximum observable return in the dataset. The DT architecture is identical to that of CtRL-Sim except the return token precedes the state token, and the returns and future states are not predicted by the decoder. **4.** *CTG++* is a reimplementation of [11], a competitive Transformer-based diffusion model for behaviour simulation.

We evaluate the following variants of CtRL-Sim: **1.** *CtRL-Sim (Base)* is the CtRL-Sim model trained on the offline RL dataset. **2.** *CtRL-Sim (No State Prediction)* is the base model trained without the state prediction task. **3.** *CtRL-Sim (Positive Tilting)* applies $\kappa_c = 10$ tilting to all components $c$ of the base model. **4.** Instead of predicting the return-to-go at each timestep, *CtRL-Sim (GT Initial Return)* uses the ground-truth initial return-to-go and, at each timestep, decrements the reward from the initial return-to-go until the episode terminates. For controllability evaluation, we evaluate on the base model and a finetuned CtRL-Sim model (*CtRL-Sim FT*). The finetuned model takes a trained base model and finetunes it on a dataset of simulated long-tail scenarios that we collect using an existing simulated collision generation method CAT [21]. This allows CtRL-Sim to be exposed to more long-tail collision scenarios during training, as the WOMD mainly contains nominal driving. We refer readers to Appendix H for details of CAT and our proposed finetuning procedure.

### 3.2 Results

Table 1 presents the multi-agent imitation results comparing the CtRL-Sim model and its variants with imitation baselines. The CtRL-Sim models perform competitively with the imitation baselines, with the CtRL-Sim (Positive Tilting) model achieving a good balance between distributional realism (2nd in JSD), reconstruction performance (2nd in FDE, ADE), common sense (Tied 1st in Collision and Offroad Rate), and efficiency (5.4× faster than CTG++). We note that while IL is faster than CtRL-Sim due to fewer tokens to decode and offers similar performance, IL is *not controllable*. Although DT (Max Return) attains equal collision and offroad rates as CtRL-Sim, the reconstruction performance is substantially worse. We further validate the importance of the future state prediction task, with CtRL-Sim (Base) outperforming CtRL-Sim (No State Prediction) across all metrics. The CtRL-Sim (Positive Tilting) model attains the best collision rate and offroad rate, demonstrating the effectiveness of exponential tilting for steering the model towards good driving behaviours.

A distinctive feature of CtRL-Sim is that it enables intuitive control over the agent behaviours through exponential tilting of the return distribution. This contrasts with DT, which, although capable of generating suboptimal behaviours by specifying low initial return-to-gos, lacks intuitive

| Adv. Method | Tilt | Reactive? | Control? | Planner Metrics | | Adversary Realism | |
| | | | | Progress (m) $\downarrow$ | Coll. w/ Adv. (%) $\uparrow$ | JSD ($\times 10^{-2}$) $\downarrow$ | Coll. Speed (m/s) $\downarrow$ |
|---|---|---|---|---|---|---|---|
| CAT | | ✗ | ✗ | 53.3 | 61.4 | 18.7 | 6.9 |
| CtRL-Sim | $-10$ | ✓ | ✓ | $57.5_{\pm 0.1}$ | $10.0_{\pm 0.5}$ | $10.8_{\pm 0.3}$ | $7.4_{\pm 0.5}$ |
| | 10 | | | $57.7_{\pm 0.1}$ | $8.7_{\pm 0.5}$ | $10.3_{\pm 0.5}$ | $8.3_{\pm 0.6}$ |
| CtRL-Sim FT | $-10$ | ✓ | ✓ | $56.1_{\pm 0.2}$ | $33.8_{\pm 1.9}$ | $17.4_{\pm 0.6}$ | $6.3_{\pm 0.2}$ |
| | 10 | | | $57.1_{\pm 0.1}$ | $18.5_{\pm 1.6}$ | $12.6_{\pm 0.7}$ | $6.1_{\pm 0.2}$ |
| | 50 | | | $57.4_{\pm 0.2}$ | $12.8_{\pm 0.2}$ | $15.6_{\pm 1.2}$ | $6.0_{\pm 0.3}$ |

Table 2: **Adversarial scenario generation results over 1000 test scenes.** We report the mean$\pm$std over 5 seeds for the CtRL-Sim models. Finetuning CtRL-Sim on CAT data improves ability to generate adversarial scenarios compared with base CtRL-Sim model. Compared with CAT, CtRL-Sim is reactive and controllable, while exhibiting better collision realism.

control due to the prerequisite knowledge about the return-to-go values and an absence of an interpretable mechanism for behaviour modulation. By contrast, the exponential tilting employed in CtRL-Sim has a clear interpretation: negative exponential tilting yields behaviours that are worse than the average behaviours learned from the dataset, while positive exponential tilting yields better-than-average behaviours. We show the results of our controllability evaluation in Figure 4. For each reward dimension $c$, we exponentially tilt $\kappa_c$ between -25 and 25 and observe how this affects the corresponding metric of interest. We also show the results of DT when conditioning on the minimum and maximum possible return. For both the base and finetuned CtRL-Sim models, we observe a relatively monotonic change in each metric of interest as the tilting coefficient is increased. As the finetuned model is exposed to collision scenarios during finetuning, it demonstrates significant improvements over the base model in generating bad driving behaviours. Specifically, at -25 tilting, the finetuned model is able to generate $2.1\times$ as many collisions and $1.8\times$ as many offroad violations as the base model. Figure 5 (and 7, 8 in Appendix J) shows qualitatively the effects of tilting.

Table 2 evaluates CtRL-Sim's ability to produce adversarial agents that collide with a data-driven planner. We evaluate on a held-out test set of two-agent interactive scenarios from the Waymo interactive dataset, where one interacting agent is controlled by the adversary and the other is controlled by the planner. We use a positively-tilted CtRL-Sim base model as our planner, due to its demonstrated ability to produce good driving behaviours in Table 1. For adversarial scenario generation, we compare the base CtRL-Sim model against the CtRL-Sim FT model. With -10 tilting applied, the finetuned model generates 238 more collisions with the planner than the base model over 1000 scenes, which we attribute to its exposure to simulated collision scenarios during finetuning. Notably, CtRL-Sim FT was finetuned in only 30 minutes on 1 NVIDIA A100-Large GPU and only 3500 CAT scenarios. This underscores CtRL-Sim's capability to flexibly incorporate data from various sources through finetuning, thereby enabling the generation of new kinds of driving behaviours. Importantly, after finetuning, CtRL-Sim FT largely retains its ability to produce good driving behaviours. This is evidenced by a 21.1 percentage point decrease in the planner's collision rate when using a +50 positively tilted finetuned model as the adversary. We also compare CtRL-Sim against a state-of-the-art collision generation method, CAT [21], which uses a motion prediction model to select plausible adversarial trajectories that overlap with the ego plan. Although CAT generates more collisions, CAT is *not controllable* as it can't control how adversarial the agents are, and CAT agents are *non-reactive* to the ego's actions as the trajectory is fixed prior to the simulation, limiting its realism. This is evidenced by a larger adversary collision speed than all finetuned CtRL-Sim models and is also validated qualitatively in the supplementary video. We further conduct a user study to confirm that CtRL-Sim adversarial scenarios are indeed more realistic than CAT adversarial scenarios, with details and results reported in Appendix I.

## 4 Related Work

Agent behaviour simulation involves modelling the behaviour of other agents in simulation, such as vehicles and pedestrians. Agent behaviour simulation methods can be categorized into rule-based

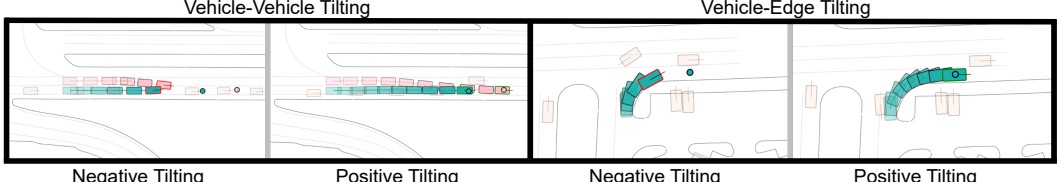

Figure 5: **Qualitative results of vehicle-vehicle and vehicle-edge tilting.** Two traffic scenes comparing positive tilting of the CtRL-Sim-controlled agent (shown in teal) with negative tilting for the same agent. Bounding boxes in red indicate traffic violations. Other agents log-replay through physics, with interacting agents in pink. Goals are marked by small circles.

and data-driven methods. Rule-based methods rely on human-specified rules to produce plausible agent behaviours, such as adhering strictly to the center of the lane [2, 3]. These methods often yield unrealistic behaviours that fail to capture the full spectrum of driving behaviours. To address these limitations, prior work has proposed learning generative models that aim to replicate agent behaviours found in real-world driving trajectory datasets [22, 23, 24, 4, 5, 25, 9]. These approaches draw inspiration from methods for the task of joint motion prediction [26, 27, 28, 29, 30]; however, it's crucial to distinguish that, unlike the open-loop nature of joint motion prediction, behaviour simulation operates closed-loop [31]. To improve the realism of the learned behaviours, other work has proposed using adversarial imitation learning [32] to minimize the behavioural discrepancy between expert and model rollouts [33, 34, 6] or RL to improve traffic rule compliance [35, 36]. While such methods demonstrate improved realism over rule-based methods, they lack the necessary control over the behaviours to enable the generation of targeted simulation scenarios for AV testing.

More recent work has proposed more controllable behaviour simulation models by learning conditional models [10, 37, 7, 38, 8, 11, 12] that enable conditioning on a high-level latent variables [10, 37], route information [7], or differentiable constraints [8, 11, 39, 12, 40]. More recently, [41] used retrieval augmented generation to generate controllable traffic scenarios. However, these methods either lack interpretable control over the generated behaviours [37] or require costly test-time optimization procedures to steer the generated behaviours, such as latent variable optimization [10], Bayesian optimization [42, 43, 7], or the simulation of expensive diffusion processes [8, 44, 11, 39, 12, 40]. In contrast, CtRL-Sim offers a more efficient alternative and learns a conditional multi-agent behaviour model that conditions on interpretable factorized returns, thereby eliminating the need for costly test-time optimization. By exponentially tilting the predicted return distribution [16] at test time, CtRL-Sim enables *efficient, interpretable, and fine-grained control* over agent behaviours while being grounded in real-world data.

## 5 Conclusion

We presented CtRL-Sim, a novel framework applying offline RL for controllable and reactive behaviour simulation. Our proposed multi-agent behaviour Transformer architecture allows CtRL-Sim to employ exponential tilting at test time to simulate a wide range of agent behaviours. We present experiments showing the effectiveness of CtRL-Sim at producing controllable and reactive behaviours, while maintaining competitive performance on the imitation task compared to baselines.

**Limitations** The learned policies produced by CtRL-Sim in its current form may make large-scale RL training on CtRL-Sim scenarios prohibitively expensive. However, we believe that further optimizations of the model and simulator such as exploring more compact models, utilizing model quantization or Flash Attention [45], reducing the planning frequency, or making the simulator GPU-accelerated can further improve the inference latency. We leave this for future work. Furthermore, CtRL-Sim enables control over the behaviours, but not the initial agent placements. We believe that a generative model that supports controlling both the initial agent placements and the agent behaviours is promising future work to explore.

**Acknowledgments**

LP and CP are supported by CIFAR under the Canada CIFAR AI Chair program and by NSERC under the Discovery Grants program.

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

| Approach | Density Estimation | Action Sampling Density |
|---|---|---|
| Decision Transformers (DTs) | $\log p_\theta(a_t\|o_t, G_t)$ | $p_\theta(a_t\|o_t, G_t)$ |
| Reward Weighted Regression (RWR) | $\exp(\eta^{-1}G_t)\log p_\theta(a_t\|o_t)$ | $p_\theta(a_t\|o_t)$ |
| Reward Conditioned Policies (RCPs) | $\log p_\theta(a_t\|o_t, G_t)p_\theta(G_t\|o_t)$ | $p_\theta(a_t\|o_t, G_t)p_\theta(G_t\|o_t)\exp(\kappa G_t - \eta(\kappa))$ |
| Reweighted Behavior Cloning (RBC) | $\log p_\theta(G_t\|o_t, a_t)p_\theta(a_t\|o_t)$ | $p_\theta(G_t\|o_t, a_t)p_\theta(a_t\|o_t)\exp(\kappa G_t - \eta(\kappa))$ |
| Implicit RL via SL (IRvS) | $\log p_\theta(a_t, G_t\|o_t)$ | $p_\theta(a_t, G_t\|o_t)\exp(\kappa G_t - \eta(\kappa))$ |
| Model-Based RCPs (MB-RCP) | $\log p_\theta(o_{t+1}\|a_t, o_t, G_t)p_\theta(a_t\|o_t, G_t)p_\theta(G_t\|o_t)$ | $p_\theta(a_t\|o_t, G_t)p_\theta(G_t\|o_t)\exp(\kappa G_t - \eta(\kappa))$ |
| RCP with Future Rollout (CtRL-Sim ) | $\log p_\theta(s_{t+1:T}\|a_t, s_t, G_t)p_\theta(a_t\|s_t, G_t)p_\theta(G_t\|s_t)$ | $p_\theta(a_t\|s_t, G_t)p_\theta(G_t\|s_t)\exp(\kappa G_t - \eta(\kappa))$ |

Table 3: Offline policy modelling approaches in prior work. We can see that methods differ in the decomposition of the joint distribution over actions and returns, with some approaches utilizing state prediction as a regularizer. We note that this table is adopted from prior work [18, 17].

## A  Offline RL Approaches

We position CtRL-Sim within the field of offline RL. Table 3 presents the different ways explored in the literature for learning policies in offline RL, and how these methods can sample return-maximizing actions at test time. Prior methods in offline RL differ in how the policy is modelled during training and how inference is performed; we refer the reader to Table 3 in the Appendix A for a breakdown of the different approaches. As CtRL-Sim employs a return-conditioned policy for controllable simulation, we briefly present a class of related methods that learn return-conditioned policies (RCPs) [15, 16, 46, 47, 48, 18, 49] for offline RL. RCPs are concerned with learning the joint distribution of actions and returns, such that action sampling is conditioned on the return distribution. Instead of modelling the return distribution, the Decision Transformer [15] conditions the learned policy on the maximum observed return, $G_{\max}$ in the dataset. That is, at inference time, the actions are sampled from $a_t \sim p_\theta(a_t|s_t, G_{\max})$. Lee et al. [16] instead propose to employ the learned return distribution, combined with exponential tilting, in order to sample high-return actions while remaining close to the empirical distribution. CtRL-Sim adopts exponential tilting of the predicted return distribution for each agent to finely control agent behaviour. Additionally, CtRL-Sim explicitly models the future sequences of states.

## B  Action Sampling Algorithm

Algorithm 2 describes the proposed action sampling procedure for controllable behaviour generation with factorized exponential tilting.

---

**Algorithm 2** The action sampling algorithm used by CtRL-Sim to allow for factorized tilting of the exhibited behaviour.

---

1: **Input:** $\{\kappa^1, \ldots, \kappa^C\}$     ▷ The specified inverse temperature for each return-to-go component.
2: **for** $c = 1$ to $C$ **do**
3:    $G_t'^c \sim p_\theta(G_t^c|s_t, s_G)\exp(\kappa^c G_t^c)$
4: **end for**
5: $a_t \sim \pi_\theta(a_t|s_t, s_G, G_t'^1, \ldots G_t'^C)$
6: **return** $a_t$

---

## C  Nocturne Physics Simulator and Offline RL Dataset

### C.1  Physics-based Nocturne Simulator

CtRL-Sim extends the Nocturne simulation environment [13]. Nocturne is a lightweight 2D driving simulator that is built on real-world driving trajectory data from the Waymo Open Motion Dataset [14]. A scene in Nocturne is represented by a set of dynamic objects – such as vehicles, pedestrians, and cyclists – and the map context, which includes lane boundaries, lane markings, traffic signs, and crosswalks. Each dynamic object is prescribed a goal state, which is defined as the final waypoint in the ground-truth trajectory from the Waymo Open Motion Dataset. If there exist missing timesteps

in the ground-truth trajectory, we re-define the goal as the waypoint immediately preceding the first missing timestep. By default, the dynamic objects track its 9 second trajectory from the Waymo Open Motion Dataset at 10 Hz. The Nocturne Simulator is originally designed for the development of RL driving policies, where the first 10 simulation steps (1s) of context is provided and the RL agent must reach the prescribed goal within the next 80 simulation steps (8s).

We extend Nocturne by integrating a physics engine based on the Box2D library for enabling realistic vehicle dynamics and vehicle collisions. We model the vehicle's dynamics using basic physics principles, where forces applied to the vehicle are translated into acceleration, influencing its speed and direction, and with frictional forces applied to simulate realistic sliding and adherence behaviors. This extension additionally ensures that an agent's acceleration, braking, and turn radius are bound by plausible limits and that vehicles can physically collide with each other. Such improvements open the possibility of more accurately simulating complex conditions, such as emergency braking maneuvers, slippery roads, and multi-vehicle collisions.

### C.2  Offline RL Dataset Collection

The actions are defined by the acceleration and steering angle and the reward function is decomposed into three components: a goal position reward, a vehicle to vehicle collision reward, and a vehicle to road edge collision reward. We confirm that the trajectory rollouts obtained by feeding Waymo scenes through the simulator attain a reasonable reconstruction of the ground-truth Waymo trajectories (see Table 1). Following Nocturne [13], we omit bicyclist and pedestrian trajectories from the Waymo Open Motion Dataset and we omit scenes containing traffic lights. This yields a training, validation, and test set containing 134150, 9678, and 2492 scenes.

For each agent, to obtain the action $a_t$ at timestep $t$, we compute the acceleration and steering value using an inverse bicycle model computed from the agent's current state in the simulator $\hat{s}_t$ and the ground-truth next state from the trajectory driving log $s_{t+1}$. We clip accleration values between -10 and 10 and steering values between -0.7 and 0.7 radians. We then execute $a_t$ with our proposed forward physics dynamics model to obtain the agent's updated state $\hat{s}_{t+1}$, and we repeat until the agent has completed the full rollout. Table 1 confirms that this approach to offline RL trajectory data collection yields a reasonable reconstruction of the ground-truth driving trajectories.

We compute rewards at each timestep, where our reward function is factored into three rewards components: a *goal position* reward, *vehicle-vehicle collision* reward, and *vehicle-road-edge collision* reward. We chose the reward functions based on the information provided by the Nocturne simulator, which includes indicators for goal success, vehicle-vehicle collisions, and vehicle-edge collisions. For each agent, their goal is set to the final state (i.e., position, heading, and velocity) of the ground-truth logged trajectory. The goal position reward is defined by:

$$R_g(s_t, s_G) = \mathbb{1}_{\text{goal achieved}}(s_t, s_G),$$

where goal achieved$(\cdot)$ is 1 if the agent ever reaches within 1 metre of the ground-truth goal, and 0 otherwise. The vehicle-vehicle collision reward is defined by:

$$R_v(s_t, \mathbb{S}_t - \{s_t\}) = -10 \times \mathbb{1}_{\text{vehicle-vehicle collision}}(s_t, \mathbb{S}_t - \{s_t\})$$
$$+ \frac{\min(\text{dist-nearest-vehicle}(s_t, \mathbb{S}_t - \{s_t\}), 15)}{15},$$

where dist-nearest-vehicle$(\cdot)$ computes the distance between the agent of interest and its nearest agent in the scene. Finally, the vehicle-road-edge collision reward is defined by:

$$R_e(s_t, m) = -10 \times \mathbb{1}_{\text{vehicle-road-edge collision}}(s_t, m) + \frac{\min(\text{dist-nearest-road-edge}(s_t, m), 5)}{5},$$

where dist-nearest-road-edge$(\cdot)$ computes the distance between the agent and the nearest road edge.

## D  Evaluation Metrics

The goal success rate is the proportion of evaluated agents across the evaluated test scenes that get within 1 metre of the ground-truth goal position at any point during the trajectory rollout. The

| Method | JSD ($\times 10^{-2}$) | | | | |
|---|---|---|---|---|---|
| | Lin. Speed | Ang. Speed | Accel. | Nearest Dist. | Meta-JSD |
| Replay-Physics* | 0.1 | 11.5 | 17.4 | 1.2 | 7.6 |
| Actions-Only [9] | 4.1 ± 0.7 | 16.8 ± 0.3 | 15.6 ± 0.5 | 5.1 ± 0.5 | 10.4 ± 0.3 |
| Imitation Learning | **1.0 ± 0.1** | 13.4 ± 0.3 | 16.9 ± 0.4 | 2.1 ± 0.2 | 8.3 ± 0.1 |
| DT (Max Return) [15] | 2.7 ± 0.1 | 13.4 ± 0.2 | 15.4 ± 0.5 | 2.2 ± 0.3 | 8.4 ± 0.1 |
| CTG++ [11] | 3.2 ± 0.9 | **11.9 ± 0.9** | **12.4 ± 0.4** | 2.2 ± 0.3 | **7.4 ± 0.2** |
| CtRL-Sim (No State Prediction) | 1.2 ± 0.1 | 13.7 ± 0.2 | 15.9 ± 0.7 | 2.0 ± 0.2 | 8.2 ± 0.2 |
| CtRL-Sim (Base) | 1.1 ± 0.2 | 13.8 ± 0.2 | 15.6 ± 0.5 | 2.0 ± 0.3 | 8.1 ± 0.2 |
| CtRL-Sim (Positive Tilting) | 1.4 ± 0.1 | 13.6 ± 0.2 | 14.8 ± 0.5 | **1.8 ± 0.2** | 7.9 ± 0.1 |
| DT* (GT Initial Return) | 1.1 ± 0.2 | 13.4 ± 0.2 | 16.8 ± 0.6 | 2.1 ± 0.2 | 8.4 ± 0.1 |
| CtRL-Sim* (GT Initial Return) | 1.1 ± 0.2 | 13.8 ± 0.3 | 15.3 ± 0.6 | 2.2 ± 0.2 | 8.1 ± 0.2 |

Table 4: Breakdown of Meta-JSD in Table 1. For each metric, the best unprivileged method is **bolded** and second-best is underlined. * denotes a privileged method requiring the ground-truth future trajectory.

final and average displacement errors are calculated for all evaluated agents across the test scenes and averaged. For a specific scene $s$, the collision rate and offroad rate of $s$ are the proportion of evaluated agents in $s$ that collide with another agent or road edge, respectively. These rates are then averaged across all tested scenes to define the overall collision and offroad rates.

We compute the Jensen Shannon Distance (JSD) between the distributions of features computed from the real and simulated rollouts. The Jensen Shannon Distance between two normalized histograms $p$ and $q$ is computed as:

$$\sqrt{\frac{D_{\text{KL}}(p||m) + D_{\text{KL}}(q||m)}{2}},$$

where $m$ is the pointwise mean of $p$ and $q$ and $D_{\text{KL}}$ is the KL-divergence. Unlike prior works that compute the Jensen Shannon Divergence [6, 7], we compute its square root – the Jensen Shannon Distance – so that values are not too close to 0. We compute the JSD over the following feature distributions: linear speed, angular speed, acceleration, and nearest distance. Since the acceleration values are discrete, for the acceleration JSD, we define one histogram bin for each valid acceleration value, yielding 21 evenly spaced bins between -10 and 10. For the linear speed histogram, we use 200 uniformly spaced bins between 0 and 30. For the angular speed JSD, we use 200 uniformly spaced bins between -50 and 50. For the nearest distance JSD, we use 200 uniformly spaced bins between 0 and 40.

## E  Individual JSD Results

In Table 4, we report the per-feature JSD results for Table 1.

## F  CtRL-Sim Training and Inference Details

**Training:** The CtRL-Sim behaviour simulation model is trained using randomly subsampled sequences of length of $H \times N \times 3$, where $H = 32$ and $N = 24$. For the actions, we discretize the acceleration and steering into 20 and 50 uniformly quantized bins, respectively, yielding 1000 action tokens. For the return-to-gos, we discretize each return-to-go component $G_t^{c,i}$ into 350 uniformly quantized bins. All agents and the map context are encoded in global frame as in [28, 9] where we center and rotate the scene on a random agent during training. The map context is represented as a set of road segments $m := \{r_l\}_{l=1}^{L}$, where each road segment is defined by a sequence of points $r_l := (p_l^1, \ldots p_l^P)$, where $L$ is the number of road segments and $P$ is the number of points per road segment. We apply a per-point MLP to the points of each road segment $r_l$. To produce road segment-level embeddings, we then apply attention-based pooling [50] on the embeddings of the points within each road segment, yielding $L$ road segment embeddings of size $d$. We select the

$L = 200$ closest lane segments within 100 metres of the centered agent as the map context, and select up to $N = 24$ closest agents within 60 metres of the centered agent as social context for the model. For each lane segment, we subsample $P = 100$ points. We use a hidden dimension size $d = 256$, where we use $E = 2$ Transformer encoder blocks and $D = 4$ Transformer decoder blocks, and we set $\alpha = \frac{1}{100}$ in the loss function. We supervise our model only on the trajectories of moving agents. We found it useful to employ *goal dropout* whereby the embeddings for $10\%$ of agent goals are randomly set to 0 to prevent the model from overrelying on the goal information. We found goal dropout useful for learning an informative map representation. The state, return, and action embeddings for the missing timesteps are set to 0. To ensure that the model is permutation equivariant to the agent ordering [28, 27], we modify the standard temporally causal mask by additionally enforcing that each agent can only attend to its own action and return-to-go tokens at the present timestep while allowing access to all agents' state tokens at the present timestep and all agents' tokens in the past timesteps. The CtRL-Sim model is trained using a linear decaying learning rate schedule from 5e-4 for 200k steps using the AdamW optimizer and a batch size of 64. At inference, we sample actions with a temperature of 1.5. The CtRL-Sim architecture comprises 8.3 million parameters that we train in 20 hours with 4 NVIDIA A100 GPUs.

**Inference:** CtRL-Sim supports scenes with an arbitrary number of agents. As CtRL-Sim is trained with up to $N = 24$ agents, when the number of CtRL-Sim-controlled agents at inference time exceeds $N = 24$, we iteratively select 24-agent subsets at each timestep for processing until all agents have been processed. We first randomly select a CtRL-Sim-controlled agent, we normalize the scene to this agent and select the 23 closest context agents to the CtRL-Sim-controlled agent to comprise the first set of 24 agents. We then iteratively continue centering on a CtRL-Sim-controlled agent *that has not been processed in the previous sets of 24 agents* and select its 23 closest agents for context until all CtRL-Sim-controlled agents have been included in a 24-agent subset. If an agent belongs to multiple 24-agent subsets, we use the model's first prediction of that agent. At inference time, the context length is set to training context length $H = 32$. At each timestep, we select $H = 32$ most recent timesteps as context and we found it useful to always center and rotate the scene on the centered agent at the oldest timestep in the context. For the first 10 timesteps (1s) of the simulated rollout, the states and actions are fixed to the ground-truth states and actions from the offline RL dataset, whereas the return-to-go is predicted at every timestep of the simulated rollout.

# G  Baseline Details

In this section, we describe the design decision of each baseline employed in our work. We note that for all models below, we scaled them in order for all models to have approximately the same number of learnable parameters as CtRL-Sim's architecture.

**Actions Only** The *actions-only* baseline is encoder-decoder architecture implemented in exactly the same way as CtRL-Sim with a few ablations. These include removal of states and returns from the decoder sequence, and no state rollout predictions. This model was inspired by [9] but differs in that the model also has access to the agents' goals.

**Imitation Learning** The *imitation learning* baseline is also based on the CtRL-Sim multi-agent behaviour simulation architecture but lacks factorized return information. It is a step better than the *actions-only* baseline since it considers the states in the decoder sequence, and this is corroborated by its improved performance on multi-agent simulation results of Table 1.

**DT** The *Decision Transformer* baseline is based on the seminal work [15]. We adopt an identical architecture to CtRL-Sim's with some minor difference based on the algorithm. One such decision is the lack of a return prediction based on states, and instead returns are chosen at inference time based on domain knowledge. Returns are the first token fed to the decoder, followed by states in order to predict actions. We make the strong argument that this is suboptimal for controllability (results in Figure 4) and does not provide intuitive mechanism for selecting the return values to target.

| Method | ADE (m) ↓ | FDE (m) ↓ | Goal Success Rate (%) ↑ | JSD ($\times10^{-2}$) ↓ | Collision (%) ↓ | Off Road (%) ↓ | Per Scene Gen. Time (s) ↓ |
|---|---|---|---|---|---|---|---|
| CTG++[†] | 1.72 | 3.97 | 41.7 | 7.7 | 6.4 | 17.4 | 44.0 |
| CTG++ (128 hidden dim) | 1.83 | 4.32 | 37.8 | 7.6 | 7.0 | 17.7 | 25.0 |
| CTG++ (100 diffusion steps) | 1.87 | 3.98 | 43.0 | 8.6 | 7.9 | 16.8 | 140.0 |

Table 5: **Ablations of CTG++ on Multi-agent simulation results over 1000 test scenes.** We report the results across all metrics of different configurations of the CTG++ baseline [11]. † indicates the original model results, with 256 hidden dimension and 50 diffusion steps at inference time.

**CTG++** The *CTG++* baseline is a diffusion model reimplementation of the recent work by [11]. We attempted to follow the architecture as closely as possible with a few minor differences. One such difference is that we diffuse over both states and actions, rather than diffusing over only actions and using an unicycle dynamics model to derive the states. We chose this approach because the underlying dynamics of the physics-enhanced Nocturne simulator is not necessarily governed by a unicycle dynamics model, and thus using a unicycle dynamics model would induce small errors in the derived states during training. We further note that we cannot replace the unicycle dynamics model with the Nocturne physics dynamics model as this forward model is not differentiable. We also condition on the present timestep and goal, to ensure fair comparison with CtRL-Sim. We note that at scene generation time, we diffuse over actions and states at a rate of 2 Hz which is consistent with the original CTG++ model. In addition, although we train with 100 diffusion steps, we run evaluations with 50 diffusion steps. As showing in Table 5, the difference in performance is insignificant. As we do not have information on the size of the network used in the original manuscript, we explored different hidden dimension sizes of the transformer architecture of the diffusion model, also shown in Table 5. A final difference is the future relative encoding. While CTG++ use the ground-truth to compute the relative encoding during training and a constant velocity model at test time, we opted to use the final historical timestep's relative encoding. We found this approach to be more stable.

## H    CAT Simulated Data Collection and CtRL-Sim Finetuning

The Waymo Open Motion dataset largely contains nominal driving scenes. To enhance control over the generation of safety-critical scenarios, we finetune CtRL-Sim on a simulated dataset of safety-critical scenarios generated by CAT [21]. CAT is a state-of-the-art collision generation method that involves fixing the agent's future trajectory to a trajectory predicted by a DenseTNT trajectory predictor. CAT searches for a trajectory that has high likelihood of colliding with the log-replay future trajectory of the ego vehicle, while having high probability under the behaviour prior (DenseTNT). For more details, we refer readers to [21]. We note that a limitation of CAT is that the agent is non-reactive to the ego as the agent's trajectory is fixed at the beginning of the simulation. Moreover, unlike CtRL-Sim, CAT does not have control over the degree to which the agent is adversarial.

To collect the simulated safety-critical dataset, we run CAT on a subset of the interactive validation split of the Waymo Open Motion Dataset, which involves two interacting agents. Following CAT, we select one of the two interacting agents to be the ego (whose trajectory is fixed to the log-replay trajectory) and the other interacting agent to be the CAT adversary. In total, we collect 3577 CAT scenarios for finetuning, of which around 60% contain ego-adversary collisions.

To encourage CtRL-Sim to learn how to generate safety-critical scenarios without forgetting how to generate good driving behaviour, we adopt a continual pre-training strategy for finetuning [51] where we randomly sample 3577 real training scenarios from the offline RL dataset in each training epoch, or a 50% replay ratio. We rewarm the learning rate to the maximum learning rate of 5e-4 over 500 steps and follow a linear decay learning rate schedule to 0 over 20 epochs. We expect that the finetuned CtRL-Sim model will be more capable of generating long-tail scenarios as it is more exposed to such scenarios during finetuning. Finetuning takes roughly 30 minutes on 1 NVIDIA A100-Large GPU.

# I Adversarial Scenario Generation User Study

We conduct a user study that contained a total of 24 participants to evaluate which method (CtRL-Sim vs. CAT) generates more plausible adversarial behaviours. We did not record any identifying information of the participants, and participants were invited on a voluntary basis. The user study contained a total of 271 paired scenarios, where each paired scenario consisted of two interacting agents, one controlled by a positively-tilted CtRL-Sim planner and the other controlled by an adversary. The scenario conditions of the paired scenarios are identical, except one scenario employed a CtRL-Sim adversarial agent and the other employed a CAT adversarial agent. The adversarial agent in each paired scenario is highlighted in pink so that users can distinguish the adversarial agent from the remaining agents in the scene. We do not identify the planner agent as we found this to detract users' attention from the behaviour of the adversarial agent. Each participant was tasked with evaluating a randomly selected set of 30 paired scenarios from the pool of 271 paired scenarios. For each paired scenario, users selected in which scenario the adversarial agent was more realistic, with the option to select "Tie" if the two scenarios were sufficently similar. Users were presented with the user interface shown in Figure 6, where we randomize the order between both videos. We use the same positively tilted CtRL-Sim planner in both of the paired scenarios. Table 6 shows the tally of the user study. The results indicate convincingly that CtRL-Sim produces more plausible adversarial behaviours than CAT based on the participants' preferences, as CtRL-Sim was preferred 54 more times than CAT over the 271 paired scenarios.

| Method | Times Preferred |
| --- | --- |
| CtRL-Sim | 123 |
| CAT | 69 |
| Tie | 79 |

Table 6: **The tally of votes for the larger study.** We show the breakdown of the votes for the conducted larger study. We observe that the results of this study are less conclusive than the pilot study.

## J Additional Qualitative Results

Figure 7 shows more qualitative examples demonstrating the effects of positive exponential tilting on each of the three reward components. In the left panels, CtRL-Sim with no tilting produces a vehicle-vehicle collision between two interacting agents at a left-turn. With positive vehicle-vehicle tilting, the CtRL-Sim-controlled agent moves more to the right-hand side of the lane to avoid the collision. In the middle panels, CtRL-Sim with no tilting produces a vehicle-edge collision as the bus pulls into the curb. With positive vehicle-edge tilting, the CtRL-Sim-controlled agent pulls into the curb at a safer distance from the curb. In the right panels, CtRL-Sim with no tilting reaches the goal. With positive goal tilting, the CtRL-Sim-controlled agent reaches the goal much faster and nearly avoids collision with the turning vehicle. In Figure 8, we show two more examples of adversarial collision scenarios generated with negative vehicle-vehicle tilting. We refer the interested reader to the supplementary video for more examples.

## K Multi-Agent Simulation Results with Higher Temperature Sampling

In Table 7, we report results from the same experiments as Table 1 except with a higher action sampling temperature, set to 1.5.

## L Fine-tuning CtRL-Sim on CtRL-Sim Scenarios

Instead of finetuning on CAT scenarios, we explore finetuning CtRL-Sim on adversarial scenarios generated by CtRL-Sim. We first collect a simulated dataset of scenes either containing a vehicle-vehicle collision or an offroad infraction. Specifically, we generate rollouts of a single agent with the negatively tilted base CtRL-Sim model where the other agents are set to log replay through physics, and we save the scenario only if the generated rollout yields a vehicle-vehicle

# Survey

**In which video does the pink vehicle behave more like a human driver? If neither video is preferred, select the "Tie" option.**

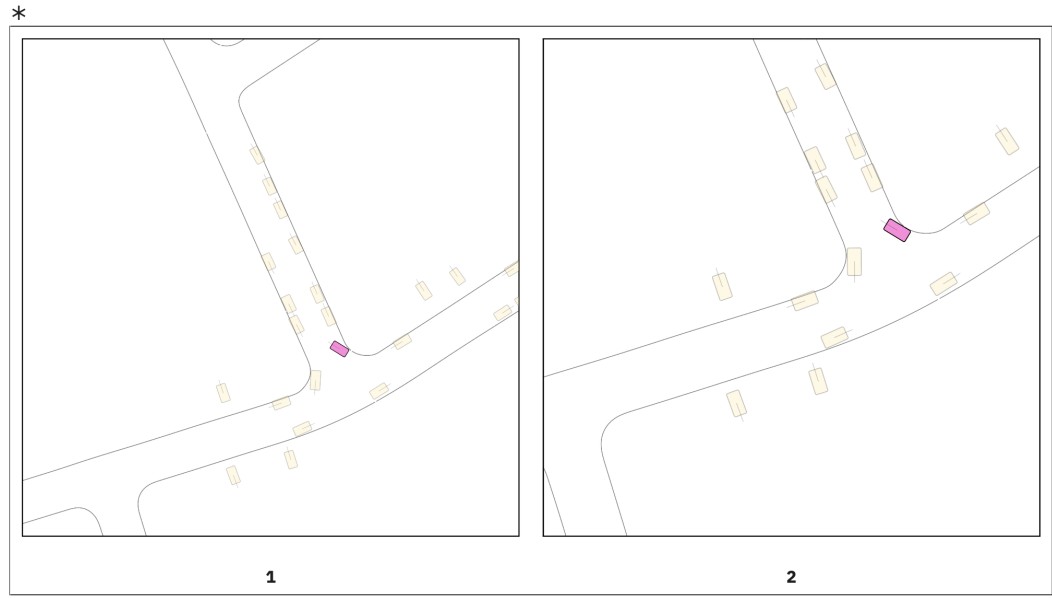

Figure 6: **User Study Example Scenario.** We show an example of a pair of scenarios along with the question users are asked to answer.

| Method | | Reconstruction | | Distributional Realism | Common Sense | |
| --- | --- | --- | --- | --- | --- | --- |
| | FDE (m) | ADE (m) | Goal Suc. Rate (%) | Meta JSD($\times 10^{-2}$) | Collision (%) | Off Road (%) |
| Replay-Physics* | 0.97 | 0.47 | 87.3 | 7.6 | 2.8 | 10.7 |
| Actions-Only [9] | $11.70 \pm 1.12$ | $4.78 \pm 0.42$ | $34.4 \pm 1.3$ | $14.3 \pm 0.3$ | $22.8 \pm 0.7$ | $29.7 \pm 1.7$ |
| Imitation Learning | $\underline{2.42 \pm 0.17}$ | $\underline{1.47 \pm 0.07}$ | $\mathbf{73.8 \pm 1.2}$ | $12.3 \pm 0.5$ | $7.3 \pm 0.6$ | $13.1 \pm 0.4$ |
| DT (Max Return) [15] | $3.25 \pm 0.17$ | $1.67 \pm 0.05$ | $60.5 \pm 1.2$ | $12.3 \pm 0.4$ | $\mathbf{6.1 \pm 0.7}$ | $\mathbf{11.6 \pm 0.3}$ |
| CtRL-Sim (No State Prediction) | $2.57 \pm 0.16$ | $1.52 \pm 0.07$ | $66.2 \pm 1.0$ | $12.3 \pm 0.3$ | $7.6 \pm 0.7$ | $13.1 \pm 0.3$ |
| CtRL-Sim (Base) | $2.49 \pm 0.10$ | $1.50 \pm 0.04$ | $\underline{67.9 \pm 1.2}$ | $\underline{12.2 \pm 0.2}$ | $7.6 \pm 0.3$ | $13.1 \pm 0.5$ |
| CtRL-Sim (Positive Tilting) | $\mathbf{2.38 \pm 0.08}$ | $\mathbf{1.44 \pm 0.03}$ | $67.2 \pm 1.0$ | $\mathbf{12.1 \pm 0.1}$ | $\underline{6.7 \pm 0.4}$ | $\underline{12.3 \pm 0.3}$ |
| DT* (GT Initial Return) | $1.94 \pm 0.07$ | $1.28 \pm 0.02$ | $73.7 \pm 1.5$ | $12.2 \pm 0.3$ | $6.6 \pm 0.4$ | $12.6 \pm 0.4$ |
| CtRL-Sim* (GT Initial Return) | $1.97 \pm 0.08$ | $1.30 \pm 0.03$ | $71.1 \pm 0.9$ | $12.2 \pm 0.1$ | $7.2 \pm 0.5$ | $13.1 \pm 0.3$ |

Table 7: **Multi-agent simulation results over 1000 test scenes with action temperature = 1.5 over 3 seeds.** This table presents the results from the same experiments as Table 1, but with an action sampling temperature of 1.5 instead of 1.0. This allows for a comparison of the impact of the temperature hyperparameter. Overall, an action sampling temperature of 1.0 yields better results.

collision or vehicle-road-edge collision. For tilting, we uniformly sample $\kappa_{\text{veh-veh}} \sim \mathcal{U}(-25, 0)$ and $\kappa_{\text{goal}} \sim \mathcal{U}(-25, 0)$ when generating vehicle-vehicle collision scenarios, and we uniformly sample $\kappa_{\text{veh-edge}} \sim \mathcal{U}(-25, 0)$ and $\kappa_{\text{goal}} \sim \mathcal{U}(-25, 0)$ when generating vehicle-road-edge collision scenarios. By additionally negatively tilting the goal, this grants the model more flexibility when generating traffic violations as the agents are not trying to reach its prescribed goal. We collect 5000 scenarios of each type of traffic violation derived from the training set, which comprises the simulated dataset of safety-critical scenarios. To encourage CtRL-Sim to learn how to generate safety-critical scenarios without forgetting how to generate good driving behaviour, we adopt the same finetuning strategy as in Appendix H, except we randomly sample 90000 real training scenarios from the offline RL dataset in each training epoch, or a 90% replay ratio. We find it useful to use a larger replay ratio when finetuning on CtRL-Sim scenarios. The controllability results are shown in Figure 9, demonstrating similar control over adversarial behaviours as the CAT-finetuned CtRL-Sim model.

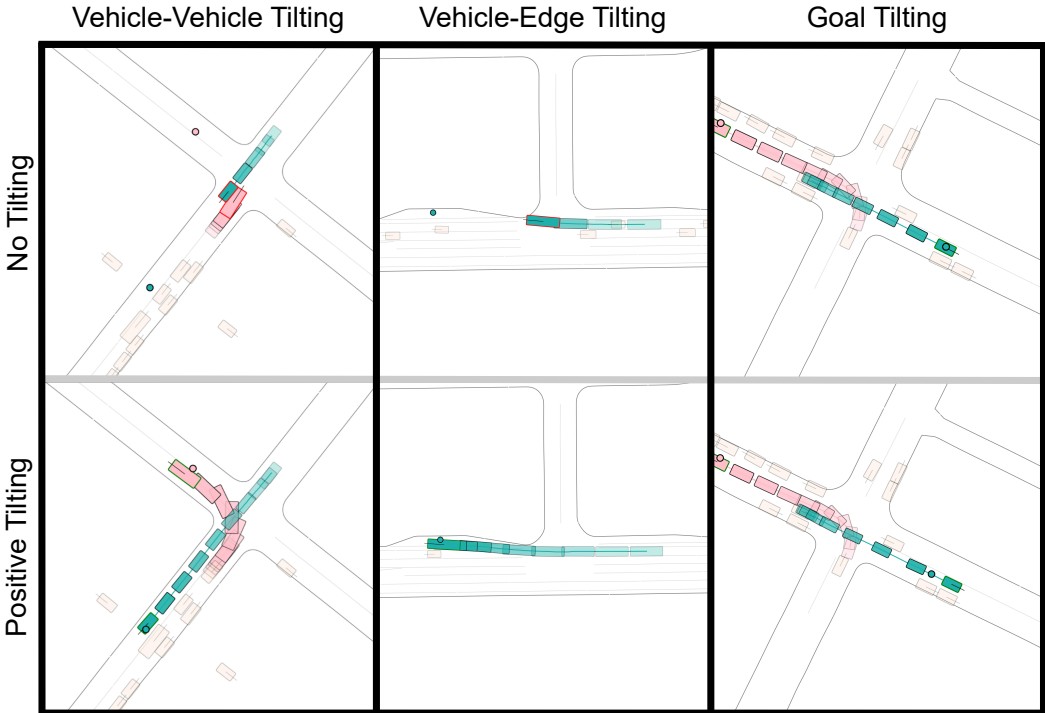

Figure 7: **Qualitative results of the effects of positive tilting.** We show the evolution of three traffic scenes with the top panels applying no exponential tilting to the CtRL-Sim-controlled agent (shown in teal) and the bottom panels applying positive tilting to the same CtRL-Sim-controlled agent. Bounding boxes outlined in red contain a traffic violation. All other agents are set to log-replay through physics, with the agent interacting with the CtRL-Sim-controlled agent denoted in pink. Goals are denoted by small circles.

| Tilting | Goal Success Rate (%) |
|---------|----------------------|
| 0 | $69.0 \pm 0.7$ |
| -10 (Late) | $64.1 \pm 1.3$ |
| 10 | $63.1 \pm 2.6$ |

(a) Goal Tilting

| Tilting | Collision Rate (%) |
|---------|-------------------|
| 0 | $14.5 \pm 1.5$ |
| -10 (Late) | $18.2 \pm 2.3$ |
| 10 | $28.9 \pm 1.8$ |

(b) Vehicle-Vehicle Tilting

| Tilting | Offroad Rate (%) |
|---------|-----------------|
| 0 | $15.2 \pm 0.9$ |
| -10 (Late) | $17.4 \pm 1.0$ |
| 10 | $21.6 \pm 0.6$ |

(c) Vehicle-Edge Tilting

Table 8: We report the results of the CtRL-Sim FT model over 1000 test scenes applying (a) Goal Tilting, (b) Vehicle-Vehicle Tilting, and (c) Vehicle-Edge Tilting. We either apply no tilting, -10 tilting, or 0 tilting for the first half of the rollout and -10 tilting for the second half of the rollout (*i.e.,* Late). We report the mean ± std over 5 seeds.

## M  Dynamic Exponential Tilting

In this section, we experiment with *dynamic exponential tilting*, where the tilting value varies depending on the simulation timestep. We specifically experiment with *late tilting* to test if tilting is still effective at later timesteps. We applied a -10 tilting to the CtRL-Sim FT model only during the last half (4 seconds) of the generated rollout. We evaluated over 5 seeds on 1000 scenes, as in Figure 4. We evaluate the tilting of each reward component independently. The results are shown in Table 8, where we report the mean±std over 5 seeds.

As expected, with late tilting, we observe a higher goal success rate and a lower collision/offroad rate compared to the model that applies negative tilting at all timesteps. Conversely, we see a lower goal success rate and a higher collision/offroad rate compared to the model with no tilting. This

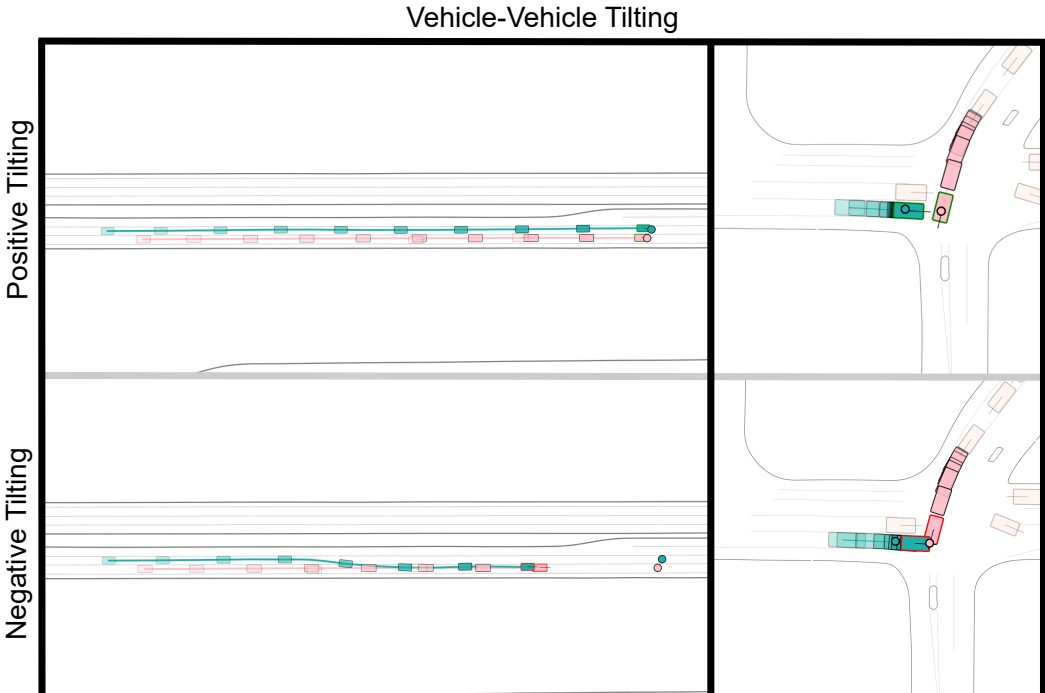

Figure 8: **Qualitative results of vehicle-vehicle tilting.** We show the evolution of two traffic scenes with the top panels applying positive exponential tilting to the CtRL-Sim-controlled agent (shown in teal) and the bottom panels applying negative tilting to the same CtRL-Sim-controlled agent. Bounding boxes outlined in red contain a traffic violation. All other agents are set to log-replay through physics, with the agent interacting with the CtRL-Sim-controlled agent denoted in pink. Goals are denoted by small circles.

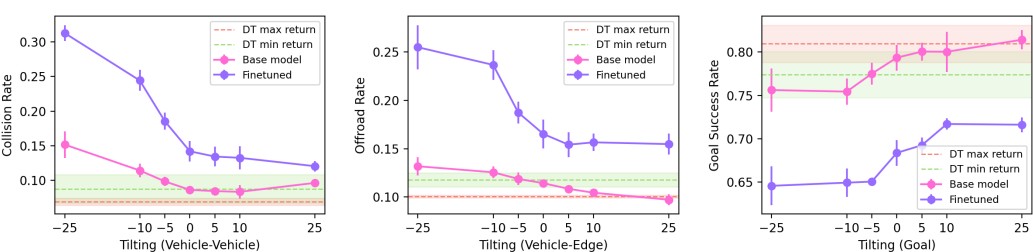

Figure 9: **Effects of exponential tilting.** Comparison of CtRL-Sim base model (magenta) and a CtRL-Sim model fine-tuned on adversarial CtRL-Sim scenarios (purple). As opposed to Figure 4, this fine-tuned model does not involve using CAT to select the adversarial scenarios. Rewards range from -25 to 25 for vehicle-vehicle collision (**left**), vehicle-edge collision (**middle**), and goal reaching (**right**). Results show smooth controllability, with fine-tuning enhancing this effect. Mean and std are reported over 5 seeds.

shows that tilting remains effective at the later timesteps. An interesting observation is that late tilting performs similarly to full tilting of the goal reward, likely because avoiding the goal can be effectively achieved within the last 4 seconds of the rollout, even if the agent was initially traveling towards the goal. On the other hand, it may be more difficult for the agent to collide with another agent or a road edge in the last 4 seconds if it hasn't planned for that in the first 4 seconds.

# N Failure Cases

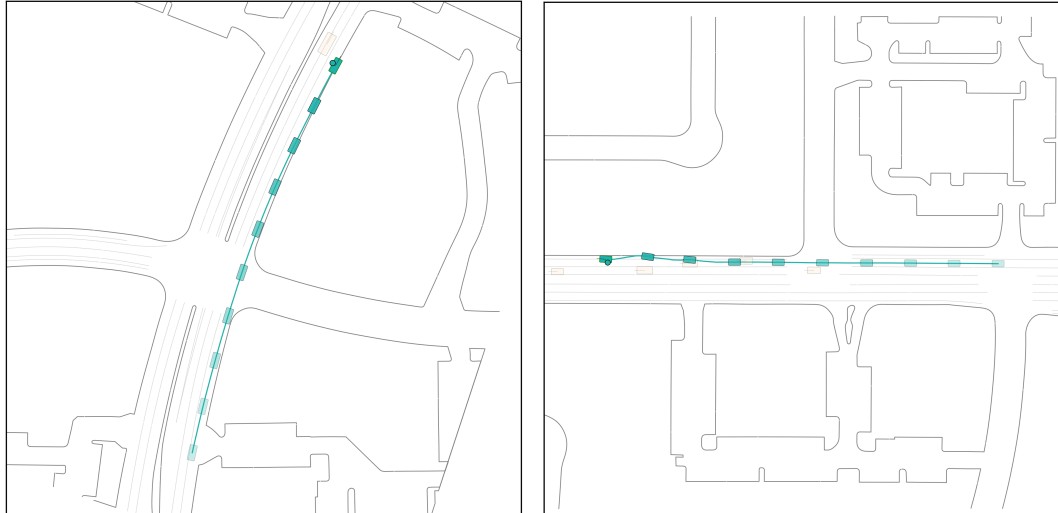

Figure 10: **Failure cases.** Visualization of two failure cases of CtRL-Sim. The agent controlled by CtRL-Sim is depicted in teal. On the left, the agent drifts off the road, likely due to imitation drift. On the right, the agent's behaviour is erratic and implausible due to extreme tilting values (-50).

In Figure 10, we visualize two failure cases generated by CtRL-Sim. In the scenario on the left, we control one agent using CtRL-Sim with zero tilting, indicated in green. However, the agent drifts off the road, exhibiting a failure mode likely caused by imitation drift—a common issue in imitation learning methods when rolled out over time. This drift likely results from the model's lack of exposure to sufficient examples of agents successfully recovering their position when approaching the edge of the road. In the scenario on the right of Figure 10, we control one agent using CtRL-Sim with -50 vehicle-vehicle tilting. The agent's behaviour is erratic and implausible, which is caused by selecting a large negative tilting value that pushes the model out of distribution. This failure mode can be mitigated by choosing tilting values that are not too extreme. We find empirically that tilting CtRL-Sim with values between -25 to 25 generally produce more plausible outputs.

