# OpenReview forum: "CtRL-Sim: Reactive and Controllable Driving Agents with Offline Reinforcement Learning"
_robot-learning.org/CoRL/2024/Conference — CoRL 2024_

### Official Review · Reviewer_KJao · 2024-07-05
**This paper proposes an offline deep RL-based methodology to increase variability of autonomous driving simulator.**

**Originality:** 2
**Technical Quality:** 3
**Clarity Of Presentation:** 4
**Potential Impact:** 2
**Recommendation:** 3
**Confidence:** 4

**Review:**

The proposed idea is well described and easy to follow. The paper provides quantitative and qualitative experimental results of the resulted agent.

Strength:
- The inverse temperature for the factorized return-to-go is an intuitive knob for users to manipulate behavior of agents at test time.
- Since the inverse temperature doesn't involve to the training process, the training algorithm remains almost as simple as the base algorithm.
- Both quantitative and qualitative analysis is provided with sufficient number of baselines to show the advantage of the proposed method.

Weakness:
- The paper doesn't provide this analysis, but to train Decision Transformer-based agent with a good coverage of variable behavior, training datasets might need to have sufficient diversity to interpolate return-to-go space. In this paper, it seems not a problem since they used an open dataset. But, it could be potentially an issue especially when the large-scale dataset is not available.
- From deep RL perspective, dynamically weighting multiple objectives is not novel. There are some papers practically doing similar things[1][2].  While I think this paper proposes a novel application in autonomous driving community, authors need to add descriptions to clarify how this paper is positioned in deep RL community.


References:
- [1] Abels, Axel, Diederik Roijers, Tom Lenaerts, Ann Nowé, and Denis Steckelmacher. "Dynamic weights in multi-objective deep reinforcement learning." In International conference on machine learning, pp. 11-20. PMLR, 2019
- [2] Yang, Runzhe, Xingyuan Sun, and Karthik Narasimhan. "A generalized algorithm for multi-objective reinforcement learning and policy adaptation." Advances in neural information processing systems 32 (2019).

**Quality Of The Limitations Section:**

1

**Questions For Rebuttal:**

My main concern is that whether the proposed idea has novelty, compared to existing multi-objective deep RL research while I acknowledge its novelty as an application work. I'd like authors to add this discussion to Section 4. It'd be even nice if authors could add additional experimental results to include baselines based on an existing multi-objetive deep RL approach.

**Robotics Focus:**

2

**Summary Of Paper:**

This paper proposes Decision Transformer-based agent with variable weightings of objectives at test time for autonomous vehicle simulation. The core part of the proposal is to introduce factorized return-to-go training and inverse temperature weights to modify agent behavior at test time.

**Summary Of Recommendation:**

From an application point of view, this paper proposes a useful methodology in autonomous driving community. But, the manuscript needs more discssion to clarify the novelty of the proposed method in deep RL community.

---

### Official Review · Reviewer_xpFT · 2024-07-06
**Review of Submission27**

**Originality:** 4
**Technical Quality:** 3
**Clarity Of Presentation:** 4
**Potential Impact:** 3
**Recommendation:** 3
**Confidence:** 5

**Review:**

**Strength:**

1.	The motivation is clear. Generating reactive and controllable scenarios is important to simulation. Existing work has not explored enough in this direction.
2.	The paper is well-written and well-organized. Most of the concepts and backgrounds are clearly stated.
3.	The analysis of experimental results is detailed and thorough.

**Weakness:**

1.	One potential issue I can think about is that the controllability of the generator is limited by the quality of the dataset. This is actually the limitation of all offline RL methods. I think the experiment of safety-critical scenario generation also supports this point. Without fine-tuning, the controllability is quite limited.
2.	The design of the reward in the proposed method seems not comprehensive enough. The three sub-rewards used seem to be particularly designed for the metrics of realism (collision and offroad).
3.	How to select the goal during generation? I think the realism heavily depends on the selection of the goal. Does it require a lot of human effort to select the goal? If so, how to generate large-scale scenarios?
4.	Some details are missing. Check my questions.
5.	Missing evaluation of the improvement of downstream tasks. Simulation is mainly used to improve driving models. It would be better if the authors could show improvement of agents using the proposed scenarios for training or evaluation.

**Quality Of The Limitations Section:**

1

**Questions For Rebuttal:**

1.	The definition of reward-to-go is not mentioned in the main context. I find the definition in the appendix, but I think the authors should at least mention what reward they used during the training. I think Figure 1 already shows all three rewards used. A justification for the selection of them will be better. The vehicle-to-vehicle and vehicle-to-edge factors seem to be carefully designed for the collision ratio and offroad ratio metrics. However, they are not sufficient conditions to show the realism of the scenario.
2.	In Figure 3, some of the agents are controlled by CtRL-Sim and others are not. Will that cause collisions between controlled agents and replayed agents? It seems the scenario is not fully reactive. If all agents are controlled by CtRL-Sim, the computation could be a burden and it is also hard to select desired goals.
3.	In Table 1, the result of Replay-Physics still has a 2.8% collision rate and 10.7% offroad rate. Does that mean the vehicle dynamics are not good enough?
4.	It is not clear what kind of initial RTG is used for CtRL-Sim (Base). I assume it is using the max RTG. Then according to the results in Table 1, CtRL-Sim ∗ (GT Initial Return) looks better than CtRL-Sim (Base). Does this mean the RTG influences the results a lot and the RTG prediction in CtRL-Sim is not accurate?
5.	According to Table 2, CtRL-Sim achieves a low collision ratio without finetuning, which might be evidence to show that the controllability is largely limited by the dataset. In addition, after fine-tuning, CtRL-Sim FT has a higher collision ratio than the model without finetune even with tilt=50. Does that mean finetuning on adversarial scenarios sacrifices the quality of the model?
6.	In DT, the input reward-to-go is decided by a linear scheduler. However, in the proposed method, it seems that the input reward-to-go is always the predicted one from the last step. Will the prediction error accumulate during the generation? Then does the tilting still work in later time steps? Some experimental results of the predicted error of RTG vs. timesteps could be helpful.
7.	It would be interesting to show more tilting examples as in Figure 5.
8.	If I understand correctly, the tilting is applied to the predicted RTG. This makes sense since some scenario initialization will not easily lead to collision and the predicted RTG provides a baseline to tune. However, this also raises a question. Should I tune the tilting value according to the specific scenario? In addition, the value may depend on the distribution of this initialization in the dataset rather than the fact. For example, some initialization may easily lead to collision but the collected samples do not contain such scenarios.

**Robotics Focus:**

3

**Summary Of Paper:**

This paper focuses on the generation of scenarios in driving simulation. Different from existing works that rely on heuristics or generative models of real-world data, the proposed method leverages return-conditioned offline reinforcement learning to efficiently generate reactive and controllable traffic agents. By controlling the desired returns for the various reward components, their method can generate a wide range of driving behaviors beyond the scope of the initial dataset. In addition, the authors also show that they can generate diverse and realistic safety-critical scenarios using adversarial generation.

**Summary Of Recommendation:**

My initial score is weak reject as I have some questions to be answered. Most of them are clarification questions so my final score will depend on the new information provided by the authors.

---

### Official Review · Reviewer_GaAn · 2024-07-22
**Good paper. Solid technical contributions. Well-written. Properly designed experiments.**

**Originality:** 3
**Technical Quality:** 4
**Clarity Of Presentation:** 4
**Potential Impact:** 3
**Recommendation:** 3
**Confidence:** 3

**Review:**

The paper proposes a return conditioned offline reinforcement learning (RL) approach, called "CtRL-Sim", to address the autonomous driving (AD) controllable simulated agents problem for the evaluation of autonomous vehicles (AV) planners. Compared to decision transformers (DT), CtRL-Sim predicts state sequences, and allows for modifying ("tilting") the rewards-to-go during the autoregressive inference process. The paper shows favorable quantitative results as well as qualitative examples to demonstrate the effectiveness of the proposed method. The paper is well-written, and easy to read with excellent clarity. Experiments are properly designed.

* Originality.
Neither the problem or the method is new. However, the paper demonstrates a practical way to apply DT style offline RL methods to AD simulated agents, which is valuable contribution to the autonomous driving research community. The method is very practical and is easily reproduced.

* Quality.
The paper is technically sound. Most of the claims are supported (except for claiming the method is "efficient") with experimental results. The method is appropriate. The paper is complete. The weaknesses of the method is not discussed in the paper. It'd be great to show some qualitative examples where the method is not performing well and explain why that's the case.

* Clarity.
Perhaps the biggest strength of the paper is its clarity. The paper is well organized and enjoyable to read.

* Significance.
The results are expected and supportive of most of the claims in the paper, however, according to the reported results, the proposed method doesn't show significant improvements over IL. I'd encourage the authors to clearly state that and explain why in the corresponding paragraph in the Experiments section. The method itself is an extension of existing methods, but it likely be used by other researcher and practitioners in the field.

* Relevance.
The paper addresses simulated agents modeling, which is an important problem in AD. The method is evaluated in the Nocturne simulator. The AD community would be interested in reading this paper.

* Limitations.
Limitations of the proposed method in this paper is not sufficiently discussed.

**Quality Of The Limitations Section:**

2

**Questions For Rebuttal:**

* Line 77, what’s the forward transition model? It is worth describing in the main text.
* Line 99: “we also found it helpful to regularize the learned policy by predicting the full sequence of future states”, why predicting the full sequence regularizes the learned policy?
* Line 152, "Discretize the actions and return-to-gos into uniformly quantized bins." have the authors considered alternative tokenization methods?
* Table 1, why is the per-scene gen. time of CtRL-Sim w/ GT initial return so high compared to w/o GT initial return? The Line 292 says “efficient” but 20s is very slow compared to imitation learning (IL), 3.4s.
* Line 196, “The DT architecture is identical to that of CtRL-Sim except the return token precedes the state token” How does the order of return token and the state token matter?
* Table 1, w/o access to privileged GT futures, IL seems to perform really well and also very fast. This result seems to suggest that the main benefit of CtRL-Sim is controllability. Is that correct?
* Table 1, is it a fair comparison between methods w/ privileged GT futures and w/ privileged GT futures? Imitative metrics (ADE, FDE and Goal Success Rate) are closely related to privileged GT future information.
* Line 240, how to choose the two agents in the two-agent interactive scenarios?
* Fig 4, it seems that FT model generally performs worse than non-FT model. Why is that?
* Limitations of the proposed method in this paper is not sufficiently discussed.
* Overall, I'd be curious how does this reactive sim agents model compare to log-playback sim agents wrt evaluating AD planners.

**Robotics Focus:**

3

**Summary Of Paper:**

The paper proposes a return conditioned offline reinforcement learning (RL) approach, called "CtRL-Sim" to address the autonomous driving (AD) controllable simulated agents problem for the evaluation of autonomous vehicles (AV) planners. Compared to decision transformers (DT), CtRL-Sim predicts state sequences, and allows for modifying ("tilting") the rewards-to-go during the autoregressive inference process. The paper shows favorable quantitative results as well as qualitative examples to demonstrate the effectiveness of the proposed method. The paper is well-written, and easy to read with excellent clarity. Experiments are properly designed.

**Summary Of Recommendation:**

I recommend a weak accept. I may raise the rating depending on how well the questions are addressed in rebuttal.

---

### Author Rebuttal · Authors · 2024-08-08

Attached:
- updated_manuscript.pdf: An updated manuscript incorporating the suggestions from all reviewers.
- rtg_error.png: Figure showing predicted RTG error vs simulation timestep (as suggested by Reviewer xpFT).

---

### Decision · Program_Chairs · 2024-09-04

**Decision:**

Accept

**Comment:**

Summary: This paper leverages return-conditioned offline reinforcement learning to generate reactive and controllable traffic agents from real-world driving data, providing fine-grained control and enabling diverse, realistic, and safety-critical scenarios in simulation.

Strength:
* The importance of generating reactive and controllable scenarios for simulation is well articulated.
* Detailed and thorough analysis of experimental results, including quantitative and qualitative examples.
* Demonstrates a practical way to apply decision transformer-style offline RL methods to autonomous driving simulation.

Weakness:
* The controllability of the generator is limited by the quality of the dataset.
* The design of the reward in the proposed method is not comprehensive enough.
* Details such as the definition of reward-to-go and goal selection during generation, are missing or insufficiently explained.
* The method's controllability is limited without fine-tuning, as evidenced by experiments with safety-critical scenario generation.

----

Post rebuttal:
The reviewers are happy with the authors' replies and have reached a consensus on accepting the paper.